# A Paradoxical Vasodilatory Nutraceutical Intervention for Prevention and Attenuation of Migraine—A Hypothetical Review

**DOI:** 10.3390/nu12082487

**Published:** 2020-08-18

**Authors:** Devahuti Rai Chaliha, Mauro Vaccarezza, Ryu Takechi, Virginie Lam, Eric Visser, Peter Drummond, John Charles Louis Mamo

**Affiliations:** 1Curtin Health Innovation Research Institute, Curtin University, Kent St., Bentley 6102, Australia; devahuti.chaliha@postgrad.curtin.edu.au (D.R.C.); mauro.vaccarezza@curtin.edu.au (M.V.); R.Takechi@curtin.edu.au (R.T.); Virginie.Lam@curtin.edu.au (V.L.); 2School of Public Health, Faculty of Health Sciences, Curtin University, Kent St., Bentley 6102, Australia; 3School of Pharmacy and Biomedical Sciences, Faculty of Health Sciences, Curtin University, Kent St., Bentley 6102, Australia; 4School of Medicine, University of Notre Dame, Fremantle 6160, Australia; eric.visser@nd.edu.au; 5College of Science, Health, Engineering and Education (SHEE), Murdoch University, Murdoch 6150, Australia; p.drummond@murdoch.edu.au

**Keywords:** aged garlic extract, calcium, L-arginine, migraine, nitric oxide, vasodilation, vasoconstriction

## Abstract

Studies suggest that migraine pain has a vascular component. The prevailing dogma is that peripheral vasoconstriction activates baroreceptors in central, large arteries. Dilatation of central vessels stimulates nociceptors and induces cortical spreading depression. Studies investigating nitric oxide (NO) donors support the indicated hypothesis that pain is amplified when acutely administered. In this review, we provide an alternate hypothesis which, if substantiated, may provide therapeutic opportunities for attenuating migraine frequency and severity. We suggest that in migraines, heightened sympathetic tone results in progressive central microvascular constriction. Suboptimal parenchymal blood flow, we suggest, activates nociceptors and triggers headache pain onset. Administration of NO donors could paradoxically promote constriction of the microvasculature as a consequence of larger upstream central artery vasodilatation. Inhibitors of NO production are reported to alleviate migraine pain. We describe how constriction of larger upstream arteries, induced by NO synthesis inhibitors, may result in a compensatory dilatory response of the microvasculature. The restoration of central capillary blood flow may be the primary mechanism for pain relief. Attenuating the propensity for central capillary constriction and promoting a more dilatory phenotype may reduce frequency and severity of migraines. Here, we propose consideration of two dietary nutraceuticals for reducing migraine risk: L-arginine and aged garlic extracts.

## 1. Introduction

Migraine is the most common and one of the costliest neurological disorders to treat [1,2]. Migraines are described to occur through a stress-induction pathway and may, in some individuals, represent a continuum of tension headaches [3]. As a global health burden, migraines have an estimated worldwide prevalence of 1.04 billion with a staggering disability-adjusted life burden of approximately 45.1 million years per annum [4]. The aetiology of migraines is not entirely clear; however, there is a body of opinion which suggests that pain in migraine is associated with vasodilatation of large cerebral arteries [5]. This hypothesis is paradoxical, as dilatation of large cerebral arteries is not suggested by studies examining net change in cerebral blood flow in migraine sufferers. Rather, the latter would suggest that, along with dilatation of large arteries, there is downstream vasoconstriction of the cerebral microcirculation, raising an alternate consideration of causation [6].

Studies implicate the trigeminovascular system in the aetiology of migraines [7], with cerebral vasodilatation being suggested to exacerbate pain by causing depolarisation of perivascular nociceptive nerve terminals and promoting vascular inflammation [8]. The primary evidence for this hypothesis comes from clinical studies investigating the acute effects of intravenous administration of nitric oxide donors such as glyceryl trinitrate (GTN). Acute administration of GTN markedly accentuates pain intensity in migraine sufferers. However, conflicting with those studies, there is accumulating evidence that decreasing oxygen tension of inspired air (via enrichment with carbon dioxide), can provide immediate relief of migraine [9,10] via an autoregulatory mechanism [11]. Carbon dioxide is a potent cerebral vasodilator, so these findings seem at odds with interpretation of the studies, considering the mechanism of pain amplification afforded by acute administration of nitric oxide (NO) donors such as GTN.

Age and sex are factors positively associated with the prevalence of migraines and, in addition, changes in cerebrovascular ‘tone’. Studies have shown that endothelial function changes with ageing [12], generally described as becoming less responsive to regulating factors and less patent (increased arterial stiffness) [13]. Women are also three times more likely to experience migraines than are men [14], which manifest around menarche and become increasingly common in their 30s and 40s. The monthly frequency of migraine also worsens during perimenopausal changes. The greater risk of migraine in women is of particular interest in the context of potential mechanisms triggering migraine. Oestrogen strongly stimulates synthesis of the potent vasodilator (NO) in arterial vascular endothelial cells, whilst concomitantly suppressing secretion of vasoconstrictive endothelin [15]. The net effect of oestrogen loss during menorrhoea would therefore notionally result in cerebro-vasoconstriction, an association that is inconsistent with the prevailing hypothesis that cerebral vasodilatation underlies headache onset during attacks of migraine. Several explanations are possible for these contradictory observations, including direct regulatory effects of oestrogen on the trigeminovascular system independent of vascular tone and/or differential triggers for migraine pain.

A paradoxical vascular axis for risk of migraine is, in part, also suggested by psychobiological studies which demonstrate that stress modulates endothelial function [16] and is associated with frequency and severity of migraines [17]. Psychosocial stress causes heightened sympathetic nervous system (SNS) and suppressed parasympathetic nervous system (PNS) responses [16], and triggers vasoconstriction via sympathetic efferent fibres that is generally proportional to the level of stress realised [16]. Indeed, stress-induced vascular constriction can be sufficient to significantly heighten blood pressure [18]. Psychological stress is also reported to be associated with vascular inflammation which can amplify vasoconstriction [19]. For example, stimulation of trigeminal nerve fibres causes plasma protein extravasation, mast cell activation, and degranulation in the dura mater [20]. The neurogenic inflammation realised is proposed to exacerbate migraines by activating trigeminal sensory fibres in the meninges [21]. It is on this basis that behavioural and cognitive therapies have been used to manage stress levels, with the intent to reduce frequency and severity of migraines [22].

### The Role of Nitric Oxide in Modulating Cerebral Vascular Tone

There is a positive association of migraine prevalence with ageing, associated with a phenotypic shift towards decreased NO production that results in impaired vasodilatory function [23]. NO is the primary endogenous vasodilator synthesised by vascular endothelial cells [24], which stimulates arterial smooth muscle cells and capillary pericytes to relax by regulating cellular ionised calcium (Ca^2+^) homeostasis. For vasoconstriction to occur, Ca^2+^ enters smooth muscle cells or pericytes through calcium ion channels and, once released from the sarcoplasmic reticulum, binds with calmodulin, resulting in myosin phosphorylation and cell constriction [25]. Endothelial-derived NO promotes vasodilatation by promoting calcium ion efflux [25] or reducing the cell contractile apparatus to calcium ions [26]. NO causes vascular relaxation via 2 pathways: a non-cGMP-dependent (cyclic guanosine monophosphate (cGMP)) stimulation of sarcoendoplasmic-reticulum calcium ATPase (SERCA) to reuptake Ca^2+^ from acting in the cytosol, and/or through soluble guanylyl cyclase (sGC)-stimulated cyclic guanosine monophosphate (cGMP) which desensitises the contractile apparatus to Ca^2+^ via L-type calcium ion channels [25]. The combined effects result in a decrease of smooth muscle cell/pericyte cytosolic Ca^2+^ ions which are recycled back into the sarcoplasmic reticulum for sequestration/storage [25]. This antagonistic process is shown in Figure 1.

## 2. A Revised Hypothesis for the Pathophysiology of Vascular-Based Migraines

Normal physiological vascular tone modulation in a healthy human brain is depicted as follows (Figure 2A). The regional microvasculature is ordinarily highly responsive to parenchymal energy requirements. With increasing energy demand, the microvascular endothelium synthesises vasodilators (NO) and releases them into the bloodstream, causing precapillary sphincters, small arterial smooth muscle cells, and capillary pericytes to vasodilate. Functional magnetic resonance imaging elegantly demonstrates marked changes in microvascular blood flow [27] with acute mental challenges. Depending on the microvascular demand, regional upstream larger vessels may also dilate in order to support changes in flow and, critically, in response to intravascular pressure. The regional intrinsic factors, which regulate microvascular tone, may be direct innervations from neuronal cells and astrocytes to endothelial cells, arteriolar smooth muscle cells, and capillary pericytes. The innervations to endothelial cells will modulate NO biosynthesis by causing intracellular distributional changes in the cytoplasmic pools of ionised calcium (Figure 1).

The prevailing dogma of vascular tone in the aetiology of migraine is illustrated in Figure 2B. Endocrine/stress pathways are thought to progressively induce peripheral vasoconstriction. Migraine patients have been reported to have exaggerated muscle-nerve sympathetic response, and to show heightened vasoconstriction during muscle contractions [28]. Peripheral microvascular vasoconstriction activates baroreceptors in central large arteries, activating endothelial nitric-oxide synthase (eNOS) and stimulating production of NO. The heightened release of NO and/or vasodilatation, per se, stimulates nociceptors and induces cortical spreading depression via K^+^ release, causing migraine aura in many subjects. Consistent with the indicated hypothesis, NOS inhibitors are found to alleviate headache pain, presumably as a consequence of inhibiting vascular dilatation.

We present an alternate hypothesis with respect to purported cerebrovascular changes in the aetiology of migraines. The hypothesis reconciles the vascular paradoxes described. We propose that as a consequence of stress-induced heightened sympathetic tone, or as a consequence of endocrine mediators (such as depletion of plasma oestrogen during menorrhoea), microvascular blood flow is compromised, not only in peripheral vascular beds, but also within the central microvasculature (Figure 3). Suboptimal central parenchymal blood flow, or indeed pseudo-hypoxia, may activate cranial nociceptors and cause headache. The acute intravenous provision of exogenous NO donors, such as GTN, will promote upstream central larger artery dilatation (Figure 3). To respond to upstream vasodilatation and critically maintain parenchymal perfusion pressure within the central microvasculature, capillaries must further constrict. The microvascular response of constriction would exacerbate the pre-existing suboptimal parenchymal flow in regional areas of high energy demand, effectively amplifying nociceptor activation and pain intensity [29]. The acute provision of NOS inhibitors to treat migraine will result in central large artery constriction, concomitant with increased heart rate. In response, downstream parenchymal microvessels would need to vasodilate in order to maintain perfusion pressure within an optimal range. The latter would restore microvascular blood flow. The restoration of flow provides nutrients and oxygen, and there is alleviation of pain.

In contrast to NO donors such as GTN, it is established that lowering blood oxygen saturation, by inhaling air enriched in carbon dioxide, will immediately result in global vasodilatation of both the peripheral and the central microvasculature (Figure 3). Our contention is that restoring central capillary perfusion, via carbon dioxide treatment protocols, would attenuate activation of nociceptors caused by insufficient perfusion of blood.

### A Nutraceutical Approach to Promote Microvascular Dilatation and Attenuate Risk for Migraine Onset and Severity

In contemporary times, with greater accessibility and affordability of non-prescription and prescription medicines, potential overuse of triptans and nonsteroidal anti-inflammatory drugs (NSAIDs) is regrettably increasingly reported [30,31]. The potential for significant contraindications and/or adverse effects of drugs used to treat migraines is therefore a significant driver to consider therapeutic opportunities with fewer potential adverse side effects. In the context of a cerebrovascular axis approach to treat migraine, there is an accumulating body of evidence that the nutraceutical agents L-arginine (2-amino-5-guanidinopentanoic acid, arginine) and aged garlic extract (AGE) may be particularly worthy of consideration.

L-arginine is a precursor for the vasodilator NO [32], and studies have shown an association between vascular tone and plasma concentrations of L-arginine [33]. In physiological conditions, the L-arginine molecule consists of a protonated α-amino group, a deprotonated α-carboxylic acid group, a 3-carbon aliphatic straight side chain, and ends with a protonated guanidino group [24]. NO production by vascular endothelial cells is enhanced by L-arginine, as NO is formed from the terminal guanidino-N atoms in L-arginine [24]. This involves two steps: mobilisation of L-arginine and conversion to NO—both of which are activated by bradykinin and the calcium ionophore A23187 [24]. In particular, it is the extracellular L-arginine which induces endothelial NO synthesis, not intracellular abundance.

Aged garlic extract (AGE, Kyolic, Allium sativum) is a capillary inflammation inhibitor [34] with the relevant active component, S-allyl-cysteine ((R)-2-amino-3-prop-2-enylsulfanylpropanoic acid; S-2-propenyl-L-cysteine; S-allyl-laevo-cysteine; S-allylcysteine) [35], which comprises the amino acid cysteine with an allyl group attached to its sulphur atom. AGE is prepared by extracting sliced raw garlic and storing it in 15–20% ethanol at room temperature for 20 months, increasing antioxidant activity [35]. There are ~1000 micrograms per gram of S-allyl-cysteine in AGE, but only 20 micrograms per gram of S-allyl-cysteine in raw garlic [36]. AGE enhances NO synthesis [37], promoting vasodilatation [38]; however, indirect positive effects on vessel diameter have been proposed because of its potent antioxidant, antithrombotic, and antiatherosclerotic effects [39]. Despite AGE demonstrating antiplatelet effects, there has been no increased bleeding risk in patients concurrently taking blood-thinning medication [40].

## 3. Vasodilatory Effects of L-arginine

Animal studies were reviewed to ascertain reported vascular effects of L-arginine. We present studies from 2009 to 2019 in Appendix A. To determine whether these effects extended to humans, we then conducted a search of completed clinical studies from 2009 to 2019 using L-arginine (Appendix A). Similar effects were noted. Moreover, only mild gastrointestinal effects were reported in humans, and those participants dropped out in only two studies [41,42].

L-arginine directly induces vasodilatation. Of note, Kharazmi et al. (2015) reported that a 0.01-M L-arginine infusion directly into a rat mesenteric vessel increased local vasodilatation [43]. Closer to the cerebrum, microdosing 30-μL of 1-mM L-arginine into the vitreous humour of a minipig increased arteriolar diameter [44]. In humans who took 1.5 g thrice daily, flow-mediated dilatation decreased at a slower rate where L-arginine was replenished [45]. With 7 g, flow-mediated dilatation and forearm blood flow increased [46,47].

We used increased blood flow as another measure of vasodilatation in humans. Decreased blood flow to brain areas led to strokes, also associated with migraines [48], so it is not surprising that vasodilatory L-arginine also improved stroke-like episodes [49]. Some studies measured reactive hyperaemia, which measures vascular dilatory response following occlusion of blood via an inflatable cuff. The surge of blood and acute increase in pressure after a brief period of vessel occlusion activates the vascular endothelium via sheer stress. Reactive hyperaemia increased in 2 studies, after giving 6.4 or 7 g of L-arginine to humans [47,50]. In NO-deficient humans, blood flow or endothelium-mediated dilatation could be re-established with 7 g thrice or 8 g twice per os (p.o.) L-arginine daily [51,52].

We expect vasodilatation to decrease blood pressure or cerebral microvascular resistance, as there is a greater flow or force of blood passing through a wider area by vessel diameter. Therefore, we looked at the effects of L-arginine on blood pressure and vascular resistance. Giving L-arginine decreased systolic and arterial blood pressures [53,54,55] and decreased vascular and precapillary resistance in rats [55]. In humans, L-arginine lowered both diastolic and systolic blood pressures [56]. In fact, there is an inverse correlation between diastolic blood pressure and blood levels of L-arginine [57]. In clinical trials, L-arginine improved hypertension [58] via increased NO formation in patients causing vasodilatation [32]. In terms of dosing, taking 3–3.3 g of L-arginine once or twice per day resulted in fewer pre-eclampsia cases later on in gestation [59,60]. Taking 4 g of L-arginine twice/thrice a day decreased blood pressure [42,61], and fewer mildly hypertensive women had to be given antihypertensives during pregnancy with this supplementation daily [62].

Studies suggest that measuring blood levels of NO, eNOS, and/or cyclic guanosine monophosphate (cGMP) may serve as surrogate markers of systemic vascular dilatation [63]. Consistent with an exogenous induction of NO, dietary supplementary L-arginine p.o. in rats and rabbits significantly increased plasma NO and cGMP [64], as well as endothelial nitric-oxide synthase (eNOS) [65]. A 500-mg/kg intraperitoneal injection (i.p.) of L-arginine increased eNOS expression/phosphorylation and NO production in rats [66]. Giving L-arginine intravenously (i.v.) or p.o. to humans increased NO concentration in the bloodstream [67].

Migraine-without-aura headaches increase with exercise [68], suggesting that greater oxygen demand drives nociception. Vasodilatation increases oxygen delivery to tissues. Therefore, we would expect a vasodilatory agent, such as L-arginine, to inhibit muscle fatigue. In healthy humans, Pahlavani et al. (2017) found that taking 2 g of L-arginine increased maximal oxygen uptake [41]. Taking 0.1 g per kg body weight or 4 g of L-arginine decreased ischaemic pain and treadmill exercise duration, respectively [42,69]. One of these studies safely used L-arginine to alleviate painful vaso-occlusive episodes in sickle-cell disease [69]. Indeed, sickle-cell disease itself is thought to derive from limited L-arginine bioavailability inducing NO-resistance [70], reflected by low L-arginine and subsequently NO serum levels during vaso-occlusion [71]. Chen et al. (2010) found that taking 5.2 g of L-arginine increased the anaerobic threshold in men aged 50–73 years old [72]. Taking 8 g of L-arginine twice a day also improved pain-free walking distance as well as endothelial vasodilatation [33].

We also used vessel elasticity as a proxy for vessel dilatation, as the amount a vessel is able to stretch perpendicularly can often determine how much it can dilate vertically. We found that peripheral vascular elasticity was increased when rats were given 300 mg/kg L-arginine [54]. However, it seemed that this vasodilatation–elasticity association is not always the case. In humans, taking 7 g of L-arginine decreased brachial artery stiffness but increased central aortic stiffness [47], as measured by decreased carotid-femoral pulse wave velocity [46]. Fahs et al. (2009) attributed this finding to increased central arterial load due to exercise, thereby requiring stiffer collagen fibres in the central arterial walls and more elastic elastin in peripheral artery walls [47]. Hence, L-arginine still causes dilatation of both central and peripheral arteries but, physiologically, central dilatation results in decreased vessel elasticity and peripheral dilatation results in increased vessel elasticity [47].

In summary, L-arginine is worth investigating as a prophylactic migraine medication, due to its potential role in the NO vasodilatory pathway [73]. Doses of 3–8 g per day appear safe in humans [32]. However, individual patient response also depends on endogenous asymmetric dimethylarginine (ADMA) levels, as it competes with (and therefore inhibits) NO synthase, which synthesises NO from L-arginine [67]. In patients with high ADMA, L-arginine seems to restore endothelial function to a vasodilating normal by increasing NO synthesis via NO synthase [67]. A study showed a linear correlation between L-arginine:ADMA ratio and pain-free walking distance [33]. Upon the addition of L-arginine, the L-arginine:ADMA ratio is normalised [51]. As with most disease conditions, earlier treatment has been recommended for better results, measured by changes in vascular function [32]. According to our proposed alternate hypothesis, dietary L-arginine should reduce blood pressure and promote tissue perfusion by promoting microvascular dilatation via enhanced biosynthesis of endothelial NO. By extension, provision of dietary L-arginine might be an effective strategy for attenuating frequency/severity of vascular-based migraines.

## 4. Vasodilatory Effects of Aged Garlic Extract

We found no studies that reported on the effects of AGE in the context of migraine. Nonetheless, we first looked at animal studies to ascertain the vascular effects of AGE, and we present studies from the last 11 years (2009–2019) in Appendix A. To see if these effects extended to humans, we then conducted a search of completed clinical studies during the same period (Appendix A). We found the following effects.

AGE can promote vasodilatation by increasing vasodilatory prostaglandins or imitating the molecule [74]. AGE has been hence shown to improve vascular function (increase vasodilatation and decrease vasoconstriction) both in humans and rats [75,76,77]. Furthermore, in rats, giving 1.2-mL/kg AGE i.p. decreased the area of a cerebral arterial infarction that would have occurred at 2 h post-infarction induction [78,79].

Temperature rebound (the temperature increase following arterial occlusion) is another measure of vascular dilatory capacity [80,81]. This is because temperature rebound occurs quicker in non-occluded arteries after arm cuff inflation obstructs blood flow temporarily [80,81]. Temperature rebound was increased in patients who took 0.25 g of AGE once daily, indicating more dilated vessels [75,82]. Larijani et al. (2013) found that asymptomatic men who took 0.3 g of AGE with 30 mg of Coenzyme Q10 every 3 months also had improved temperature rebound after 1 year [83].

As stated above, we expect vasodilatation to decrease blood pressure or cerebral microvascular resistance. AGE produced intra-abdominally lowered blood pressure and vascular resistance in rats [77], while daily 0.48–0.96 g of AGE p.o. reduced systolic blood pressure in humans [84]. In humans, central blood pressure, central pulse pressure, mean arterial pressure, and augmentation pressure improved with 1.2 g of AGE p.o. once daily [40,85]. Ried et al. (2013) found that AGE inhibited hypertension dose-dependently in humans [86].

Again, we used biochemical markers in the bloodstream to reflect smooth muscle cell vasodilatory action. In rats, AGE increased heart NO, citrulline (an eNOS by-product), and NOx (nitrite and nitrate) [77]. In humans, AGE can cause vasodilatory effects directly, as it dose-dependently inhibits vasoconstrictive endothelin 1, also controlled by NO levels [87]. Homocysteine is known to cause endothelial dysfunction directly by stimulating endothelial inflammation and by inhibiting vasodilatory NO synthesis [88]. AGE has also been shown to inhibit homocysteine [82].

As before, we used vessel elasticity as a proxy for vessel dilatation. Breithaupt-Grogler et al. (1997) found that chronic intake of garlic powder slowed age-related aortic stiffness, suggesting that garlic per se protects vessel wall elasticity [89]. This can be measured via carotid-radial pulse wave velocity (distance of pulse wave travelled from carotid to radial artery divided by time taken for that travel) [83]. Other studies showed that pulse wave velocity and arterial stiffness decreased in people who took 0.3–1.2 g of AGE daily, measured by aortic pulse wave velocity, calculated from the brachial and radial blood pressures [40,83,85]. Since the pulse wave travelled the same distance in more time, this result suggested greater vessel elasticity. So slower pulse wave velocity corresponds with lower blood pressure and increased vascular elasticity.

In summary, the body of evidence suggests that AGE could be a suitable candidate for vascular-induced migraine therapy; however, this would need to be tested, preferably through randomised controlled trials. As seen here, garlic extracts have been investigated widely as potential treatments for cardiovascular diseases [90]. Moreover, garlic-derived therapeutics have minimal adverse effects, with the major ones being limited to halitosis and bromhidrosis [91]. Halitosis applies when the raw form of garlic is used [92], whereas AGE is odourless [35]. Potential side effects of AGE include nausea and vomiting [92]. Due to its fibrinolytic property, bleeding events can potentially occur [93], so its dose has to be monitored [94]. Only 3 studies mentioned adverse side effects of AGE [40,84,86]. However, these gastrointestinal symptoms were minor, short-lived, and readily managed by the subjects [40,84,85,86]. One study even reported improved digestion [40]. According to our proposed alternate hypothesis, dietary AGE should reduce blood pressure and promote tissue perfusion by promoting microvascular dilatation by inhibiting vasoconstrictive factors. By extension, provision of dietary AGE might be an effective strategy for attenuating frequency/severity of migraines.

## 5. Pharmacological Treatment of Migraines by Modifying Cerebrovascular Tone

The reinterpretation of literature and presented hypothesis suggest that preventing or attenuating chronic microvascular constriction may reduce the frequency or severity of frequent episodic or chronic migraines. We are not suggesting this approach in the context of acute treatment. Of the currently available pharmacotherapies, no agents are generated on the basis of the alternate hypothesis provided. However, some do have vasomodulatory effects. Pharmacotherapies used to treat migraines are reviewed elsewhere [95], but we have included a synoptic table of currently used common approaches to treat migraines, shown in Appendix A.

Of the agents routinely used to treat migraines, and in some instances tension headaches, the mechanisms of action are not fully established [96,97]. Nonetheless, there are nine classes of agents with vasodilatory properties, and three drug classes reporting both vasodilatory and vasoconstrictive effects. In addition, five classes of drugs serve as vasoconstrictors. It is beyond the scope of this hypothesis-generating review to consider each individual agent; suffice to say that there are no robust studies available that explore microvascular perfusion, central to the hypothesis proposition presented.

Collectively, the primary mechanism by which preventative migraine pharmacotherapies elicit vasomodulating effects is principally by regulating serotonin or noradrenaline pathways, calcitonin gene-related peptide (CGRP), calcium channel blockers, anticonvulsants, and membrane stabilisers. Serotoninergic receptors tend to predominate in the cerebrovasculature and noradrenergic receptors in the peripheral vasculature, suggesting ordinarily a central physiological predisposition/sensitivity to a vasodilatory phenotype [98].

In some instances, medication overuse—particularly of the nonsteroidal anti-inflammatory drugs (NSAIDs) and triptans [30,31]—can paradoxically exacerbate migraine frequency and severity. Thorlund et al. (2016) hypothesised that repeated medication use increases neuronal plasticity, in turn, causing a lower threshold to migraine headache progression [30]. In addition, vasoconstrictors such as triptans cannot be used for patients with disorders involving restricted blood supply to tissues, such as cardiovascular disorders [99]. These include ischaemic heart disease, stroke, hypertension, and hemiplegic/basilar migraine [100]. Collectively, there are many potential adverse effects of these commonly prescribed synthetic drugs to treat migraines [101], and the cardiovascular contraindications may be a significant shortcoming.

## 6. Conclusions

Given the positive action on vasodilatation exerted by L-arginine and AGE in the above cardiovascular studies, we surmise that these agents should be investigated in future randomised controlled trials for prevention of migraines induced by cerebral microvascular constriction. To assess and confirm regional vasoconstriction of cerebral arterioles and capillaries, we recommend imaging of the cerebral microvasculature directly using functional imaging techniques [27]. L-arginine is also present as a minor component in AGE [102]. Although new treatments and preventions are being explored for migraine treatment [103,104], L-arginine and AGE have not been clinically considered. One limitation is that presently, no studies investigated the effects of L-arginine and AGE in the context of gender or ageing. These variables are established risk factors for headaches, concomitant with decreased endogenous biosynthesis of NO. In this review, we did not address the effects of L-arginine and AGE on inflammation or oxidative damage, lipids, or thrombotic activity, which can also affect vessel diameter. However, all studies showed an alleviating effect of both L-arginine and AGE in all 3 aspects: decreased inflammation and oxidative damage markers, decreased triglycerides and low-density lipoproteins, increased high-density lipoproteins, decreased atherosclerosis, and decreased thrombotic events. The fact that neither L-arginine nor AGE have been tried in randomised controlled trials, specifically for the prevention of migraine, is a current shortcoming. Reported well-tolerated doses of L-arginine and AGE, considered for cardiovascular disease risk, serve as a useful guide to explore the putative efficacy of L-arginine and AGE for prevention/treatment of vascular-associated migraine.

## Figures and Tables

**Figure 1 nutrients-12-02487-f001:**
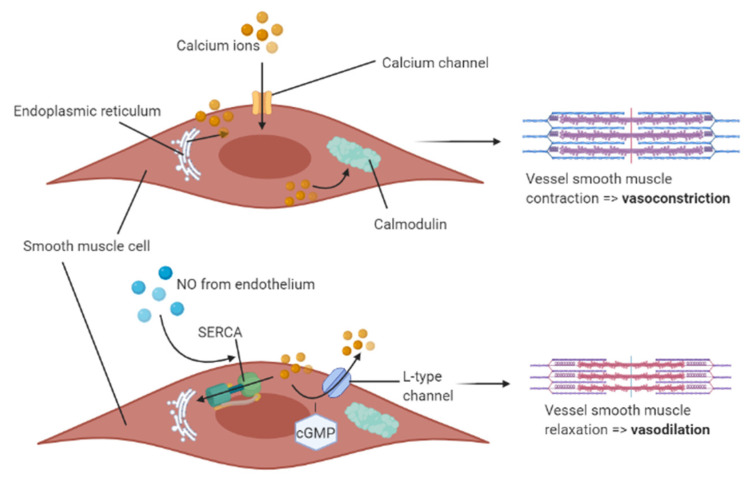
Antagonistic mechanism of NO-mediated vasodilatation and Ca^2+^-mediated vasoconstriction in the smooth muscle cell layer of blood vessels. Calcium ions bind to calmodulin in the cell cytosol, which triggers constriction of the smooth muscle cell lining the vessel. The endothelium secretes NO which prevents this pathway via two different mechanisms in the cell cytosol: SERCA-mediated and cGMP-mediated pathways. NO = nitric oxide; SERCA = sarcoendoplasmic-reticulum calcium ATPase; cGMP = cyclic guanosine monophosphate. Created on BioRender.com (BioRender).

**Figure 2 nutrients-12-02487-f002:**
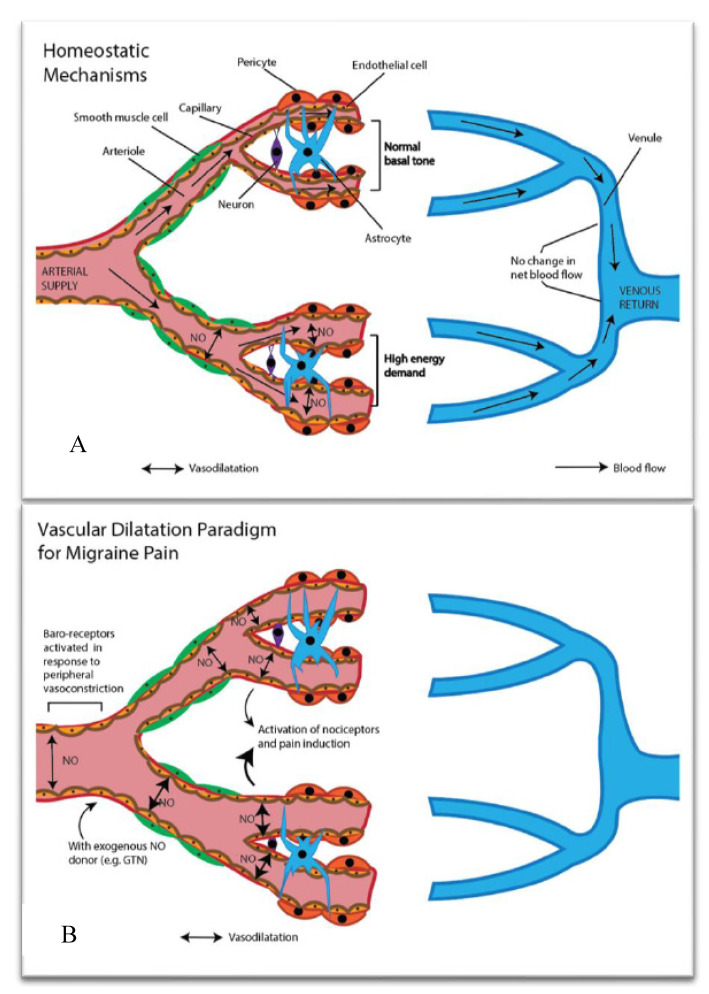
Homeostatic microvascular tone modulation and the prevailing microvascular dogma in the migraine literature. (**A**) Focal brain parenchymal changes in energy demand triggers microvascular vasodilatation, lowering resistance and increasing capillary blood flow (provision of energy substrate). Capillary dilatation is mediated through intrinsic pathway(s), including direct astrocytic modulation of pericytes, and astrocytic and neuronal innervation of endothelial cells to stimulate endothelial nitric-oxide synthase (eNOS), resulting in upregulation of synthesis of the potent vasodilator NO. Parenchymal changes in microvascular flow may be several-fold in response to regional energy demand; however, net cerebral blood flow remains essentially unchanged. “Regional” demand may result in upstream vasodilatation of the arterioles, in order to adequately supply larger regions of tissue during periods of heightened energy demand, and to maintain hydrostatic arterial pressure. (**B**) Exaggerated sympathetic tone results in peripheral vasoconstriction—for example, in the head and neck muscles. Vasomotor sympathetic-induced peripheral vasoconstriction activates baroreceptors in central large arteries, by activating eNOS and stimulating production of NO. Exaggerated NO stimulates nociceptors and induces cortical spreading depression via K^+^ release, causing migraine aura. NOS are purported to alleviate headache pain by inhibiting attenuation of vasodilatation. GTN = glyceryl trinitrate.

**Figure 3 nutrients-12-02487-f003:**
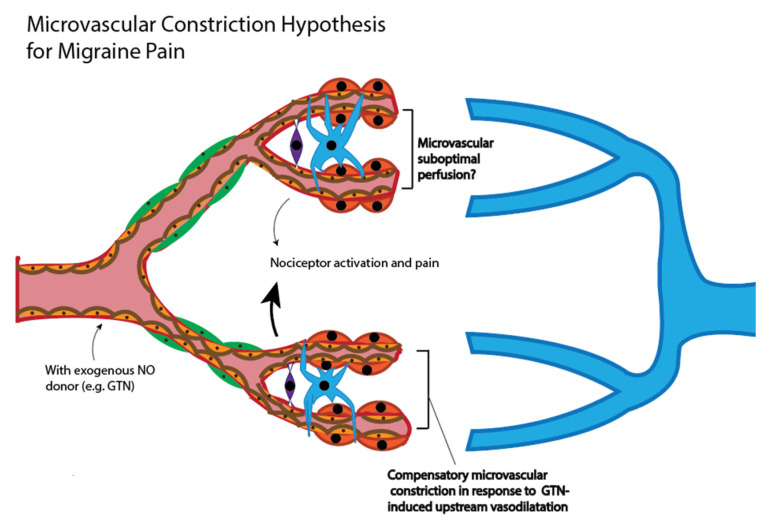
Alternate hypothesis for vascular changes in migraine. Stress-induced heightened sympathetic tone results in a chronically constrictive microvascular phenotype, both centrally and within the periphery. Suboptimal central parenchymal blood flow (potential hypoxia) could activate nociceptors and cause headache pain onset. Traditional NO donors (such as GTN) principally modulate upstream larger central arteries, and would only exacerbate downstream vasoconstriction. Provision of eNOS inhibitors will reduce blood flow to the brain through larger arteries, requiring compensatory downstream dilatation of capillaries. The latter restores the provision of nutrients and oxygen.

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
