# Peer review of "A Paradoxical Vasodilatory Nutraceutical Intervention for Prevention and Attenuation of Migraine—A Hypothetical Review"

_nutrients, 2020, doi:10.3390/nu12082487_

Round 1
Reviewer 1 Report
It is interesting to have a discussion on L-arginine as the sickle cell field is looking into this as a possible prevention for sickle cell crises and pain. May be important to cite some of these studies with Dr. Claudia Morris' team at Emory.
Line 345-346 - May need to change this sentence "In summary, the body of evidence suggests that AGE is a suitable candidate for vascular- induced migraine therapy" as there aren't mentioned RCT's or other evidence to support this statement, though one could say that the body of evidence encourages future studies to be done on L-arginine.
Starting at Line 358 sentence: "Given the positive action on vasodilatation exerted by L-arginine and AGE, we surmise that these agents should be considered for prevention of migraine induced by cerebral microvascular constriction." should also be changed as this review only supports future studies and consideration of l-arginine for prevention of migraine.
Limitations should also mention the lack of RCT trial of L-arginine for prevention of migraine specifically.
Author Response
Q1. It is interesting to have a discussion on L-arginine, as the sickle-cell field is looking into this as a possible prevention for sickle-cell crises and pain. May be important to cite some of these studies with Dr. Claudia Morris’ team at Emory.
RESPONSE: We thank the reviewer for her/his suggestion. One of the proposed studies had already been cited (reference 77), but we have now specified the sickle-cell context and included more of those studies (references 78-79). This change is along Lines 307-310.
Q2. Line 345-346 – may need to change this sentence, “In summary, the body of evidence suggests that AGE is a suitable candidate for vascular-induced migraine therapy, as there aren’t mentioned RCTs or other evidence to support this statement
RESPONSE: We have changed this to, “In summary, the body of evidence suggests that AGE could be a suitable candidate for vascular-induced migraine therapy; however, this would need to be tested, preferably through randomised controlled trials.” This is now shown along Lines 383-385.
Q3. Starting at Line 358, the sentence, “Given the positive action on vasodilatation exerted by L-arginine and AGE, we surmise that these agents should be considered for prevention of migraine induced by cerebral microvascular constriction” should also be changed, as this review only supports future studies and consideration of L-arginine for prevention of migraine.
RESPONSE: We have changed this to, “Given the positive action on vasodilatation exerted by L-arginine and AGE in the above cardiovascular studies, we surmise that these agents should be investigated in future randomised controlled trials for prevention of migraine induced by cerebral microvascular constriction”. This can be seen along Lines 435-437.
Q4. Limitations should also mention the lack of RCT trials of L-arginine for prevention of migraine specifically.
RESPONSE: We have added, “The fact that neither L-arginine nor AGE have been tried in randomised controlled trials, specifically for the prevention of migraine, is a current shortcoming.” along Lines 449-451.
Reviewer 2 Report
Dear Authors-
Chaliha et al. Is can interesting review on providing an alternate hypothesis with having hope to create horizon to new therapeutic opportunities for attenuating migraine frequency and severity. Review is very well written and having justifiable references.
Here are few of my comments:
Title should be rephrased to… “ A Paradoxical Vasodilatory Nutraceutical Intervention for Prevention and Attenuation of Migraine- A Hypothetical Review”
Point 2.A onwards it should go under discussion section.
A table with all different approach will make it easy for readers to understand the flow and findings of the manuscript.
Thanks
Author Response
Reviewer 1:
Q1. The title should be rephrased to “A Paradoxical Vasodilatory Nutraceutical Intervention for Prevention and Attenuation of Migraine – a Hypothetical Review”
RESPONSE: We thank the reviewer for this positive suggestion, and have re-titled.
Q2. Point 2.A onwards should go under the discussion section
RESPONSE: We assume the reviewer meant point 2.1 onwards. We have moved point 2.1 to the end of discussion immediately preceding the conclusion. The references have been reordered to reflect this change.
Q3. A table with all the different approaches to treat migraine will make it easy for readers to understand the flow and findings of the manuscript.
RESPONSE: We have added a synoptic table of common alternate approaches to treat migraine and included as supplementary Table S5, referring to it along Lines 405-406 of the main text.
Round 2
Reviewer 2 Report
Dear Authors
I appreciate your changes.
Good to go from my side.
Thank you